# The effect of COVID-19 on public hospital revenues in Iran: An interrupted time-series analysis

Masoud Behzadifar[1]*, Afshin Aalipour[1], Mohammad Kehsvari[2], Banafsheh Darvishi Teli[2], Mahboubeh Khaton Ghanbari[3], Hasan Abolghasem Gorji[3], Alaeddin Sheikhi[2], Samad Azari[4], Mohammad Heydarian[5], Seyed Jafar Ehsanzadeh[6], Jude Dzevela Kong[7], Maryam Ahadi[8], Nicola Luigi Bragazzi[7]

1 Social Determinants of Health Research Center, Lorestan University of Medical Sciences, Khorramabad, Iran, 2 Vice Chancellor Treatment, Lorestan University of Medical Sciences, Khorramabad, Iran, 3 Health Management Research Institute, Iran University of Medical Sciences, Tehran, Iran, 4 Hospital Management Research Center, Iran University of Medical Sciences, Tehran, Iran, 5 Faculty of Medicine, Department of Radiology, Lorestan University of Medical Sciences, Khorramabad, Iran, 6 English Language Department, School of Health Management and Information Sciences, Iran University of Medical Sciences, Tehran, Iran, 7 Laboratory for Industrial and Applied Mathematics (LIAM), Department of Mathematics and Statistics, York University, Toronto, Canada, 8 Faculty of Medicine, Department Emergency Medicine, Lorestan University of Medical Sciences, Khorramabad, Iran

* masoudbehzadifar@gmail.com

**Data Availability Statement:** Data are available upon request to the Corresponding Author and the Ethics Committee of the Lorestan University of Medical Sciences (email: research@lums.ac.ir; tel:

## Abstract

### Background

The "Coronavirus Disease 2019" (COVID-19) pandemic has become a major challenge for all healthcare systems worldwide, and besides generating a high toll of deaths, it has caused economic losses. Hospitals have played a key role in providing services to patients and the volume of hospital activities has been refocused on COVID-19 patients. Other activities have been limited/repurposed or even suspended and hospitals have been operating with reduced capacity. With the decrease in non-COVID-19 activities, their financial system and sustainability have been threatened, with hospitals facing shortage of financial resources. The aim of this study was to investigate the effects of COVID-19 on the revenues of public hospitals in Lorestan province in western Iran, as a case study.

### Method

In this quasi-experimental study, we conducted the interrupted time series analysis to evaluate COVID-19 induced changes in monthly revenues of 18 public hospitals, from April 2018 to August 2021, in Lorestan, Iran. In doing so, public hospitals report their earnings to the University of Medical Sciences monthly; then, we collected this data through the finance office.

### Results

Due to COVID-19, the revenues of public hospitals experienced an average monthly decrease of $172,636 thousand (P-value = 0.01232). For about 13 months, the trend of

00986633120172); since they may contain
potentially sensitive information and are owned by
a third-party organization (Lorestan University of
Medical Sciences, Iran).

**Funding:** The author(s) received no financial
support for the research, authorship, and/or
publication of this article.

**Competing interests:** All authors declare that there
are no conflicts of interest.

**Abbreviations:** MOHME, Ministry of Health and
Medical Education; ITSA, Interrupted time series
analysis; OLS, Ordinary least squares; OOP, Out-of-
pocket.

declining hospital revenues continued. However, after February 2021, a relatively stable
increase could be observed, with patient admission and elective surgeries restrictions being
lifted. The average monthly income of hospitals increased by $83,574 thousand.

## Conclusion

COVID-19 has reduced the revenues of public hospitals, which have faced many problems
due to the high costs they have incurred. During the crisis, lack of adequate fundings can
damage healthcare service delivery, and policymakers should allocate resources to prevent
potential shocks.

## Introduction/Background

The still ongoing "Coronavirus Disease 2019" (COVID-19) has become a major challenge for
all healthcare systems worldwide, and, besides generating a high toll of deaths, it has caused
extensive economic damage and losses [1]. All social, political, cultural and economic activities
have been affected both nationally and internationally [2]. Despite widespread efforts to con-
trol COVID-19, unfortunately, many people have lost their lives and measures and interven-
tions to prevent and control the disease are still being enforced, with only a few countries
getting back to the normal [3]. Since the beginning of the outbreak, countries around the
world have adopted various policies and programs to combat the disease, including quaran-
tine, social distancing, lockdown, closure of schools and universities, and restrictions of air,
land, and sea borders [4,5].

All actors and parts of the health system in different countries are involved to prevent, con-
trol and reduce mortality induced by COVID-19 [6]. Since the beginning of the COVID-19,
hospitals have played a very important role in providing services to patients and the volume of
hospital activities and healthcare provisions delivered has been significantly refocused on
COVID-19 patients [7]. Due to the need of these patients to receive services, many activities of
other hospitals have been limited, repurposed, or even suspended and, as such, they operate
with reduced capacity. Non-critical medical services and elective surgeries have been generally
postponed [8]. Due to the rapid spread of the disease and the increase in infections, the capac-
ity of some parts of the hospital, such as the intensive care unit, is being saturated or is reaching
a critical threshold [9]. Hospitals have changed some of their care and management protocols
to deal with this situation, and, for instance, hospital managers have closed outpatient wards
and suspended some hospital activities [10]. On the other hand, with the reduction of these
activities, the financial system of the hospitals and their sustainability are seriously threatened
and the hospitals may face a shortage of their revenues and financial resources [11]. Hospitals
need adequate economic funding to provide the needed healthcare services, and to ensure con-
tinuity of care, and, especially in critical situations such as the current COVID-19, new finan-
cial resources are needed. Health decision-makers and policymakers must pay attention to the
financial resources of hospitals to avoid reducing the provision and quality of services [12].

In February 2020, the Ministry of Health and Medical Education (MOHME) in Iran pub-
licly announced the emergence of COVID-19 in Iran when the first cases of the disease were
observed [4]. According to the MOHME report, as of January 1, 2022, out of a population of
about 84 million in Iran, 61,954,033 people were infected with COVID-19, among whom
131,639 lost their lives due to the COVID infection. Regarding vaccination, a population of

about 60 million received their first dose of the COVID-19 vaccine, 51 million received the second dose, and also 8 million received the third dose of the vaccine [13].

The MOHME, in collaboration with other organizations and ministries, implemented various policies and programs to prevent and control the COVID-19 pandemic [14]. Due to the high prevalence rate of the disease, and the influx of large numbers of people, hospitals have been providing full capacity services, especially to COVID-19 patients [15].

In the Iranian health system, services are provided in both the public and private sectors. All primary health care is provided free of charge by government centers [16]. In public hospitals, all services are provided based on government tariffs as well as patients' insurance. In these hospitals, the government has tried to keep the out-of-pocket (OOP) population low [17]. Also, private hospitals provide services based on free tariffs. The services provided in public and private hospitals are more or less the same, and people can use the services of private hospitals according to their interests [18]. The most important difference between public and private hospitals is the costs that people pay; in private hospitals in Iran, the cost of receiving services is considerably high [19].

COVID-19 has had a huge impact on the world economy, and the health financial sector has been affected and disrupted by the disease. Therefore, many governments have made extensive investments in the healthcare sector, allocating important financial resources to counteract or at least mitigate the reduction in service delivery and meet the needs of individuals [20]. In Iran, too, the government has sought to address the shortcomings of COVID-19 by injecting resources [4]. Although economic sanctions have made things difficult, the Iranian government is working to ensure that the health sector, and especially hospitals, do not suffer from a lack of funding [14]. The government injected financial aid into hospitals during the pandemic. In this regard, at the beginning of the pandemic, there was a shortage of personal protective equipment in hospitals; thus, MOHME purchased the equipment and made it available to hospitals [21]. More funding was also injected to high-prevalence provinces, and hospitals suffering from shortages of ICU beds. Meanwhile, funds were allocated for the purchase of oxygen generators and the drugs needed to fight the disease. In addition, funding was allocated to medical universities to be spent according to local priorities [22].

A large part of the healthcare services and equipment that hospitals pay for their activities is based on their revenues, and any reduction in revenues can, therefore, affect the provision of services. Studying the economic impact of COVID-19 can help health policymakers and decision-makers allocate resources and provide the funding they need and come up with plans to address revenue cuts if needed, in an evidence- and data-driven fashion. The aim of this study was to investigate the effect of COVID-19 on the revenues of public hospitals in Lorestan province in western Iran, as a case study.

## Method

### Ethics declarations

The ethics committee of the Lorestan University of Medical Sciences approved this study (approval code no. IR.LUMS. REC.1399.083).

**Setting.**   Lorestan province is located in western Iran and its population is about 1,700,000 people. Lorestan University of Medical Sciences (LUMS) is the main custodian of health services in the province and the primary health care centers and public hospitals are under its supervision and carry out their activities. There are 18 public hospitals in the province.

**Study design.**   This is a quasi-experimental study performed using interrupted time series analysis (ITSA). We performed an ITSA to evaluate changes in monthly revenue of the public hospitals due to COVID-19. Policymakers are interested in evaluating the impact of different

policies in the health sector and, depending on the effects, decide whether to continue or not their implementation [23]. ITSA is widely used to monitor and track the effectiveness of both clinical and non-clinical interventions and can help provide evidence for decision-making [24]. ITSA examines a continuous time series of observations on an outcome repeatedly over time [25]. At ITSA, we seek to understand the effects of implementing a policy or intervention on a topic by comparing before and after, and this can lead to a better understanding of the policy [26]. In health-related crises such as COVID-19, ITSA can assess the consequences of implementing different policies in these situations and help managers make effective decisions [15].

**Data collection.**    The monthly revenue was collected from April 2018 to August 2021. The revenue of 18 public hospitals was provided by LUMS financial center; two researchers (MB, MA) were referred to this data center in order to collect the data. Also, public hospitals collect their revenues monthly and report them to the LUMS financial center. In all public hospitals, the revenue department is responsible for collecting all costs, as well as determining activities related to insurance, drug price control, activities related to increasing incomes, providing hospital deductions, and OOP patients.

**Statistical analysis.**    We used a segmented regression model for ITSA [27]. The Newey-West approach was used for estimating the best analytical approach [28]. We conducted several diagnostic and sensitivity assessments to assess the robustness of our findings [29]. The ordinary least squares (OLS) regression model with a time series specification was utilized to check for serially correlated errors using the Durbin-Watson test. Also, we used visualization of the residuals from the OLS regression and plots of the autocorrelation and partial autocorrelation graphs [25]. The R software Version 3.6.1 (London, UK) was used for all statistical analyses. $P$-value $< 0.05$ was considered significant.

## Results

At the beginning of the pandemic, the average revenue of public hospitals was $460,311 thousand which began to decline with the emergence of COVID-19 in Iran. Fig 1 shows that after the onset of COVID-19 and the provision of health services under the new conditions, the general revenue of public hospitals has been associated with an average monthly decrease of $172,636 thousand. Immediately after the emergence of COVID-19, we observed a statistically significant change in hospital revenue. For about 13 months, the trend of declining hospital revenues has continued; however, after February 2021, we observed a relatively stable increase.

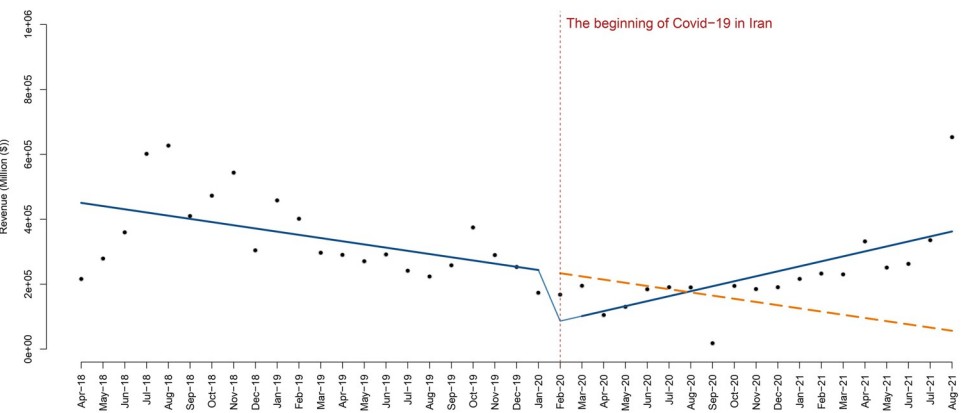

**Fig 1. The temporal trend of the revenue rate for public hospitals.**

**Table 1. The coefficients of the segmented regression model after Covid-19 in the public hospitals.**

| Variable | Value | P-Value | Lower of CI (95%) | Upper of CI (95%) |
|---|---|---|---|---|
| Intercept | 460311 | 0.000 | 367319.47 | 553302.285 |
| Time | -9850 | 0.00769 | -16930.65 | -2770.155 |
| Level | -172636 | 0.01232 | -305558.89 | -39712.465 |
| Trend | 25186 | 0.000 | 13871.47 | 36499.516 |

This decrease in income was statistically significant (P-value = 0.01232). The findings from the segmented regression model shows in Table 1.

### Return to normal conditions in hospitals

From February 2021, patient admission and selective surgeries restrictions were lifted. To assess the impact of this policy, we conducted an ITSA and analyzed the data from August 2020 to August 2021. As time passes, revenues increased, at the beginning of the hospitals' activities, the average income was $101,565 thousand. Then the average income of hospitals has experienced a monthly increase of $83,574 thousand (Fig 2). The findings from the segmented regression model after returning to normal conditions in hospitals are shown in Table 2. The autocorrelation and partial autocorrelation plots of the residuals are also reported in Fig 3A (after COVID-19 in the public hospitals) and 3B (rate after returning to normal condition in hospitals). To ensure the stability of the findings in different stages of data analysis, we conducted a sensitivity analysis and were able to confirm the stability of the findings.

### Discussion

Findings of our study revealed that after the emergence of COVID-19 in Iran, revenues of public hospitals in Lorestan province have dramatically and significantly decreased.

At the onset of COVID-19 and with the increasing prevalence of the disease, many people refused to go to hospitals for fear of infection and illness. Even if they needed to receive services in hospitals, they did not seek for medical advice, and the number of outpatient visits decreased dramatically. For better management of patients with COVID-19, selected hospital wards and units were allocated to these patients, but nevertheless, fear of the disease prevailed in the community. In many countries, hospitals had no choice but to provide more beds for patients with COVID-19, which resulted in fewer beds available for non-COVID-19 patients and decreased hospital revenues [30,31].

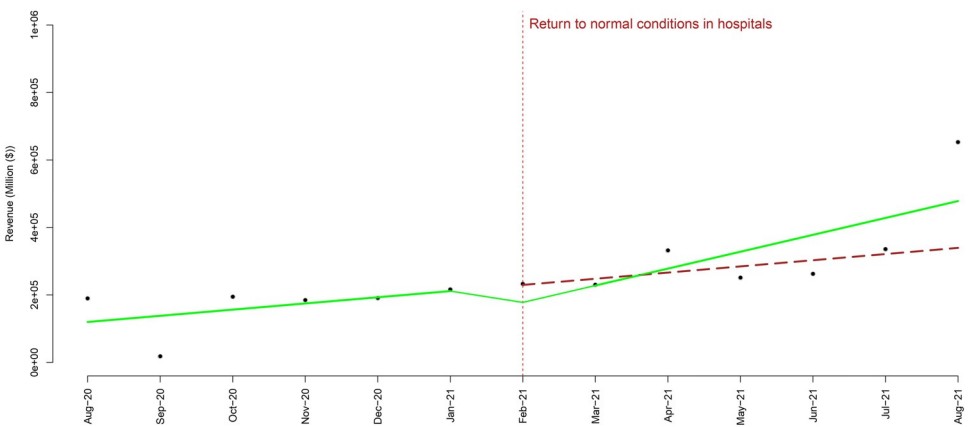

**Fig 2. The temporal trend of the revenue rate after return to normal conditions in hospitals.**

**Table 2. The coefficients of the segmented regression model after return to normal conditions in hospitals.**

| Variable | Value | P-Value | Lower of CI (95%) | Upper of CI (95%) |
|---|---|---|---|---|
| Intercept | 101565 | 0.291 | -103400.41 | 306529.49 |
| Time | 18301 | 0.452 | -34329.68 | 70930.60 |
| Level | 83574 | 0.460 | 62553.80 | 96406.16 |
| Trend | 31767 | 0.312 | 35323.84 | 98857.22 |

Kazempour-Dizaji's study showed that COVID-19 decreased hospital revenues by 9%, which is consistent with our findings (32). Khullar's study showed that the reduction in outpatient visits, due to fears of COVID-19, has been effective in reducing hospital service capacity [10].

Due to declining referrals and a sharp reduction in hospital activities, hospital managers were forced to reduce outpatient admissions, and even in some hospitals, these services were completely discontinued. Also, elective and non-emergency surgeries were completely stopped. Because of the need for more medical specialists in most hospitals to manage patients with COVID-19 and because of the possibility of transmitting the infection to non-Covid-19 patients, they were reluctant to undergo elective surgery. People who had elective surgery did not go to hospitals because of the lack of these services, and as a result, many revenues related to these services in hospitals decreased.

Decreased admissions and elective surgeries have been cited in studies as one of the reasons for the decline in revenue, and one of the main reasons for this change in revenue was MOHME's decision to turn hospitals into special centers for the treatment of COVID-19 patients and virtually no other services [32]. In the study by Bai et al., reducing hospital visits and canceling elective surgeries caused by COVID-19 decreased hospital revenues and is consistent with our findings [33]. Hospitals are unable to pay for their own expenses, especially the salaries of their staff, and face many financial problems.

On the other hand, some hospital managers, due to lack of manpower and their fatigue, reduced the admission of unnecessary patients [34]. Even in the early stages of the disease, hospitals that were not COVID-19 dedicated centers reduced patient admissions to prevent transmission of the disease to their staff. Other studies have pointed to such reasons and called for policymakers to address hospital problems, especially financial challenges [30].

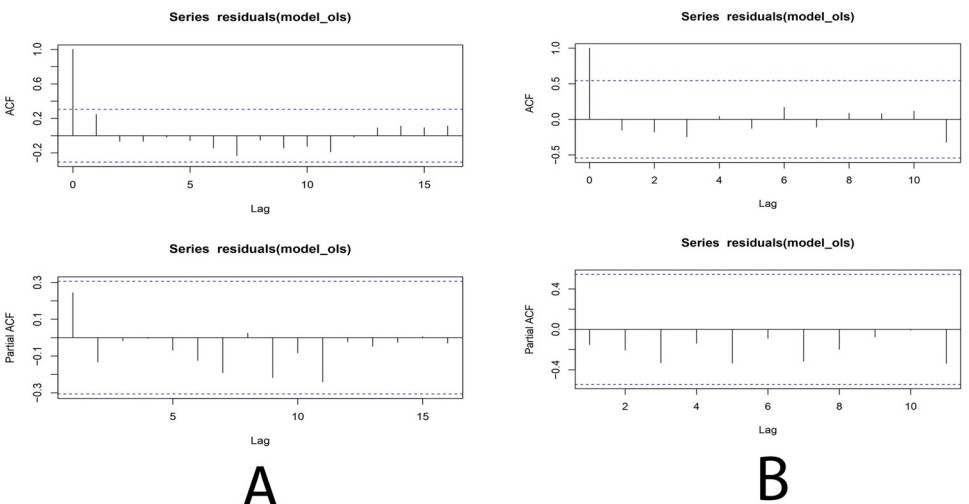

**Fig 3. Autocorrelation and partial autocorrelation function.**

The Government of Iran, in order to effectively prevent and control COVID-19, started implementing a policy of social distancing on March 27th, 2020. It also imposed severe restrictions on the presence of members of the community in various activities. Many people, for fear of fines and uncertainty about the implementation of social distance in hospitals, was one of the reasons for the decline in hospitalization for services [15].

Early in the onset of the disease in Iran, there was no specific treatment protocol for patients in hospitals, and MOHME stated that all costs of the disease were free of charge and that patients did not incur any out-of-pocket payments. The insurance companies strongly opposed this decision and did not accept the medical services provided to the patients, and the hospitals had to cope with many insurance deductions [34].

Our findings show that after 13 months, and from March 2021 on, the trend of increasing revenues of public hospitals in Lorestan province began. With the experience that hospitals gained over the months in connection with COVID-19, new policies and programs were adopted to generate revenue for public hospitals and to prevent bankruptcy. All public hospitals set up special wards and units for COVID-19 patients. Meanwhile, all hospitals began to get back to normal and started admitting different patients, and the admission restrictions were also lifted. Policies shifted toward accepting all recipients of services (either COVID-19 or non-COVID-19 patients), and the scope of hospital activities became the same as prior to COVID-19 outbreak. This policy change resulted in an increase in patients, with activities almost returned to normal.

The terrible fear existing from the beginning of the COVID-19 outbreak slowly declined, and many patients gradually came to the conclusion that it would be safe to visit hospitals while wearing masks and observing social distance. Alongside, doctors started admitting patients due to declining revenues, and hospitals increased selective surgery.

Another reason for the increase in revenues in public hospitals was the increase in safety equipment in hospitals. Hospital staff used safety equipment to provide services more safely to the public. However, their lack of safety equipment and facilities made them suffer for months, and they could not provide services to many people as usual for fear of transmitting the disease.

There is no credible scientific evidence for treating COVID-19. At present, and in recent months, due to the use of some drugs such as Remdesivir and Favipiravir for the treatment of patients, as well as the increased speed of the COVID-19 vaccination roll-out program, and the clarification of patients' treatment protocols, insurers have covered the provided services. Also, insurance reimbursements and cooperation of insurers with hospitals increased revenues.

Despite the fact that the range of activities and services provided by three private hospitals in Lorestan province is limited, and due to patients' fear of being infected with COVID-19, some patients still visited these hospitals to receive services.

## Limitations

This study has some limitations. Due to the lack of accurate infrastructure and facilities in collecting data on all revenues and expenditures, access to revenue breakdown in terms of single components was not possible; thus, we were unable to analyze all revenue components. Meanwhile, there was not enough information to analyze the role of each hospital ward and its impact on revenues. Also, we did not have access to the revenues of private hospitals. A comparison of public and private hospitals on their revenues could help understand the behavioral pattern of patients receiving services in these hospitals. In addition, cultural differences and conditions in which services are provided in hospitals in Lorestan province can be different from other provinces.

## Conclusion

The findings of this study demonstrate that COVID-19 has dramatically and significantly reduced the revenues of public hospitals, which have faced with many problems due to the high costs they have incurred. As an emerging pathogen, with destructive effects on the economy of the whole world, COVID-19 has led to staggering costs on the health sector. Many hospitals were unable to cope with COVID-19 due to lack of financial resources and declining revenues. To prevent the decline in the quality of services in hospitals, governments had to inject new financial resources into hospitals. In times of crisis, lack of adequate fundings in the health sector can cause irreparable damage to healthcare service delivery; thus, policymakers should allocate adequate funding in order to prevent potential shocks from a pandemic like COVID-19.

## Supporting information

**S1 Data.**
(XLSX)

## Author Contributions

**Conceptualization:** Masoud Behzadifar, Mohammad Kehsvari, Banafsheh Darvishi Teli.

**Data curation:** Masoud Behzadifar, Afshin Aalipour.

**Formal analysis:** Masoud Behzadifar, Jude Dzevela Kong, Nicola Luigi Bragazzi.

**Investigation:** Masoud Behzadifar, Mahboubeh Khaton Ghanbari, Hasan Abolghasem Gorji.

**Methodology:** Masoud Behzadifar, Samad Azari.

**Project administration:** Masoud Behzadifar, Alaeddin Sheikhi, Mohammad Heydarian, Seyed Jafar Ehsanzadeh.

**Resources:** Masoud Behzadifar, Maryam Ahadi.

**Software:** Masoud Behzadifar.

**Supervision:** Masoud Behzadifar, Mohammad Heydarian.

**Validation:** Masoud Behzadifar.

**Writing – original draft:** Masoud Behzadifar, Mohammad Kehsvari, Mahboubeh Khaton Ghanbari.

**Writing – review & editing:** Masoud Behzadifar, Nicola Luigi Bragazzi.

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
