## [Decision Letter · Decision Letter 0]

31 Dec 2021

PONE-D-21-30747The effects of the COVID-19 pandemic on public hospital revenues in Lorestan, Iran: insights from an interrupted time-series analysisPLOS ONE

Dear Dr. Behzadifar,

Thank you for submitting your manuscript to PLOS ONE. After careful consideration, we feel that it has merit but does not fully meet PLOS ONE’s publication criteria as it currently stands. Therefore, we invite you to submit a revised version of the manuscript that addresses the points raised during the review process.

We look forward to receiving your revised manuscript.

Kind regards,

Edris Hasanpoor

Academic Editor

PLOS ONE

Journal Requirements:

Reviewers' comments:

Reviewer's Responses to Questions

**Comments to the Author**

1. Is the manuscript technically sound, and do the data support the conclusions?

Reviewer #1: Yes

Reviewer #2: Yes

2. Has the statistical analysis been performed appropriately and rigorously? 

Reviewer #1: Yes

Reviewer #2: Yes

3. Have the authors made all data underlying the findings in their manuscript fully available?

Reviewer #1: Yes

Reviewer #2: Yes

4. Is the manuscript presented in an intelligible fashion and written in standard English?

Reviewer #1: Yes

Reviewer #2: Yes

5. Review Comments to the Author

Reviewer #1: Abstract

1. Please explain more about data collection in the method section.

2. Please delete million in the ($172,636 million) and replace with (thousand).

3. Please delete the word (dramatically) in the conclusion section.

4. Please replace (covid-19, hospital, health financing, revenue, interrupted time series analysis, health policy, Iran) with old keywords.

Main text:

1. Please update the incidence and mortality rate in the introduction section.

2. Please more explain the difference between services related to the public and private hospitals in Iran.

3. Please mention ethical approval in the method section.

4. Please explain sensitivity analysis in the results section.

5. Figure 3 is low quality, please revise and promote the quality.

6. Please explain more about limitations in the discussion section.

7. Please explain more about tables in the text in the results section.

Reviewer #2: A) The delete the (pandemic, Lorestan and insights from an) in the Title of the manuscript.

B) I think the method section can be improved in the abstract section.

C) What was the role of Iranian government assistance to hospitals?. Add sentences in the introduction section.

D) Explain how to collect hospital revenue in the method section.

6. PLOS authors have the option to publish the peer review history of their article (what does this mean?). If published, this will include your full peer review and any attached files.

Reviewer #1: No

Reviewer #2: No

---

## [Author Response · Author response to Decision Letter 0]

11 Feb 2022

Reviewer #1: Abstract

1. Please explain more about data collection in the method section.

Thanks for your comment. We added.

2. Please delete million in the ($172,636 million) and replace with (thousand).

Thanks for your comment. We added. 

3. Please delete the word (dramatically) in the conclusion section.

Thanks for your comment. Done.

4. Please replace (covid-19, hospital, health financing, revenue, interrupted time series analysis, health policy, Iran) with old keywords.

Thanks for your comment. Done.

Main text:

1. Please update the incidence and mortality rate in the introduction section.

Thanks for your comment. We added.

2. Please more explain the difference between services related to the public and private hospitals in Iran.

Thanks for your comment. Done.

3. Please mention ethical approval in the method section.

Thanks for your comment. Done.

4. Please explain sensitivity analysis in the results section.

Thanks for your comment. Done.

5. Figure 3 is low quality, please revise and promote the quality. 

Thanks for your comment. We added a high-quality figure. 

6. Please explain more about limitations in the discussion section.

Thanks for your comment. Done.

7. Please explain more about tables in the text in the results section.

Thanks for your comment. Done.

Reviewer #2:

A) The delete the (pandemic, Lorestan and insights from an) in the Title of the manuscript.

Thanks for your comment. Done.

B) I think the method section can be improved in the abstract section.

Thanks for your comment. Done.

C) What was the role of Iranian government assistance to hospitals?. Add sentences in the introduction section.

Thanks for your comment. Done.

D) Explain how to collect hospital revenue in the method section.

Thanks for your comment. Done.

---

## [Decision Letter · Decision Letter 1]

21 Mar 2022

The effect of COVID-19 on public hospital revenues in Iran: An interrupted time-series analysis

PONE-D-21-30747R1

Dear Dr. Behzadifar

We’re pleased to inform you that your manuscript has been judged scientifically suitable for publication and will be formally accepted for publication once it meets all outstanding technical requirements. Within one week, you’ll receive an e-mail detailing the required amendments. When these have been addressed, you’ll receive a formal acceptance letter and your manuscript will be scheduled for publication.

Kind regards,

Edris Hasanpoor

Academic Editor

PLOS ONE

---

## [Editor Report · Acceptance letter]

23 Mar 2022

PONE-D-21-30747R1 

The effect of COVID-19 on public hospital revenues in Iran: An interrupted time-series analysis 

Dear Dr. Behzadifar:

I'm pleased to inform you that your manuscript has been deemed suitable for publication in PLOS ONE. Congratulations! Your manuscript is now with our production department. 

Kind regards, 

on behalf of

Dr. Edris Hasanpoor 

Academic Editor

PLOS ONE